# Investigation of Aggregation and Disaggregation of Self-Assembling Nano-Sized Clusters Consisting of Individual Iron Oxide Nanoparticles upon Interaction with HEWL Protein Molecules

**DOI:** 10.3390/nano12223960

**Published:** 2022-11-10

**Authors:** Ruslan M. Sarimov, Egor I. Nagaev, Tatiana A. Matveyeva, Vladimir N. Binhi, Dmitriy E. Burmistrov, Dmitriy A. Serov, Maxim E. Astashev, Alexander V. Simakin, Oleg V. Uvarov, Venera V. Khabatova, Arthur G. Akopdzhanov, Nicolai L. Schimanowskii, Sergey V. Gudkov

**Affiliations:** 1Prokhorov General Physics Institute of the Russian Academy of Sciences (GPI RAS), 119991 Moscow, Russia; 2Russian National Pirogov Research Medical University, ul. Ostrovityanova 1, 117997 Moscow, Russia

**Keywords:** trisodium citrate–coated iron oxide nanoparticles, hen egg-white lysozyme, aggregation, fibroblast cell viability, toxicity

## Abstract

In this paper, iron oxide nanoparticles coated with trisodium citrate were obtained. Nanoparticles self-assembling stable clusters were ~10 and 50–80 nm in size, consisting of NPs 3 nm in size. The stability was controlled by using multi-angle dynamic light scattering and the zeta potential, which was −32 ± 2 mV. Clusters from TSC-IONPs can be destroyed when interacting with a hen egg-white lysozyme. After the destruction of the nanoparticles and proteins, aggregates are formed quickly, within 5–10 min. Their sizes depend on the concentration of the lysozyme and nanoparticles and can reach micron sizes. It is shown that individual protein molecules can be isolated from the formed aggregates under shaking. Such aggregation was observed by several methods: multi-angle dynamic light scattering, optical absorption, fluorescence spectroscopy, TEM, and optical microscopy. It is important to note that the concentrations of NPs at which the protein aggregation took place were also toxic to cells. There was a sharp decrease in the survival of mouse fibroblasts (Fe concentration ~75–100 μM), while the ratio of apoptotic to all dead cells increased. Additionally, at low concentrations of NPs, an increase in cell size was observed.

## 1. Introduction

Iron oxide nanoparticles (IONPs) are used in many areas of medicine: bioimaging, biosensors, drug delivery, tumor hyperthermia, and antimicrobial and antibacterial agents [1]. However, the use of IONPs is accompanied by great difficulties. Firstly, such NPs are unstable, and they form clusters over time. The aggregation process can be more intense in the living body, where nanoparticles interact with proteins [2]. As a result, in the human body, such NPs can clog blood vessels [3]. The second important difficulty is the toxicity of IONPs. Due to the Fenton reaction, IONPs can generate reactive oxygen species [4], bind to DNA [5], and therefore be toxic. Thus, the coating of IONPs with various shells solves two problems: the stabilization of nanoparticles in solution and the reduction of toxicity.

Surfactants and charged molecules are often used as shells for NPs. Trisodium citrate is one of the most popular compounds for IONP shells [6,7,8]. Sodium citrate is selected as a shell for NPs because it is relatively safe for live cells, and it has been used in medicine as an anticoagulant for more than a hundred years [9]. The trisodium citrate shell is often used to attach other compounds to NPs, such as lipase [8] or curcumin [7]. In the above example, IONPs coated with trisodium citrate (TSC-IONPs) in combination with curcumin were used for anti-cancer therapy.

TSC-IONPs are also used not only to deliver active compounds to the target point in the body but also for other tasks. TSC-IONPs are used to precipitate various proteins, for example, for the removal of toxic metal ions from an aqueous medium [10], for the removal of toxic dye [11], or as a coagulant for the *Leucaena leucocephala* protein [6]. Another study found that TSC-IONPs can be used to inhibit the formation of amyloid fibrils from lysozyme [12].

NPs in biological tissues often bind biopolymers, especially proteins, while forming a “protein crown”. In this case, protein-coated NPs can significantly change their initial properties [13]. The parameters of the protein crown itself depend on the particle size and surface properties of the NPs [14,15]. In the case of particles coated with sodium citrate, this problem is even more aggravated, because it is known that citrate-coated NPs bind well with proteins [16].

Another problem is associated with the use of the magnetic properties of NPs. It is known that nanoparticles smaller than 20 nm are superparamagnetic, which is convenient for biomedical applications since they have low residual magnetization without the application of an external magnetic field [17]. However, such NPs are extremely unstable and tend to aggregate [18]. The use of sodium citrate makes it possible to stabilize the aggregation of particles, at least until the formation of long-lived clusters. However, it is not known how stable these clusters will be when interacting with proteins. Especially with positively charged proteins, since sodium citrate has a negative charge.

In this study, a technology was created for obtaining self-assembling clusters, stable in aqueous solutions, consisting of individual iron oxide nanoparticles coated with citrate anions. The paper aimed to study the aggregation and disaggregation of self-assembling nanosized clusters consisting of individual TSC-IONPs upon interaction with HEWL protein molecules.

## 2. Materials and Methods

Materials used included hen egg-white lysozyme (>20,000 U/mg, A-3711, Applichem GmbH, Darmstadt, Germany), *Micrococcus lysodeikticus* (lyophilized cells, ATCC No. Aldrich, St. Louis, MI, USA), PBS (A9162, ITW Panreac, Barcelona, Spain), and tri-Sodium Citrate 2-hydrate, (TSC, 141655.1211, ITW Panreac, Barcelona, Spain). The water used for the experiments was produced by distillation and deionization to a resistivity of ~18 MΩ/sm.

### 2.1. Method of Producing Trisodium Citrate–Coated Iron Oxide Nanoparticles

Trisodium citrate–coated iron oxide nanoparticles (TSC-IONPs)—Fe_3_O_4_ (FeO·Fe_2_O_3_)—were obtained by chemical deposition of oxide with ammonium hydrate from an aqueous solution of a mixture of trivalent and divalent iron chloride salts, based on the technique described in previous publications [19,20].

The initial mass ratio of trivalent and divalent iron chlorides was chosen as 2:1, with a slight shift in the concentration ratio compared to the strict molar ratio [20]. The FeCl_2_·4H_2_O in a mass of 3 g was dissolved in 12.5 mL of water and 6 g of FeCl_3_·6H_2_O in 12.5 mL of water; each solution was intensively mixed separately for 15 min, at a temperature of 27 °C. The combined solution was stirred for 1 h at a selected speed of 800 rpm.

An essential feature of the formation of nano-objects obtained by this method is the determining role in the transition of oversaturated solution of ions of the initial salts from a non-equilibrium structural state to an equilibrium state corresponding to the condition of the formation of nanoparticle nuclei, as well as the time of this process. The optimal time and mode of reducing agent supply are experimentally determined in two stages of 12.5 mL of ammonium hydrate for 1 min per 25 mL of a mixture of initial salts. An increase in the rate of addition of ammonium hydrate led to an intensive clusterization process. The decrease in speed led to a significant increase in the size of the germs. The amount of ammonia is determined by the condition of complete interaction between components of the reaction.

The optimal way to control the formation of nanoparticles is to analyze the dynamics of changes in the pH of the solution. However, the determining role is played not by the pH value itself at a certain stage of synthesis, but by the dynamics of its change [20].

The citric acid (C_6_H_8_O_7_) was used as a stabilizer of nanoparticle nuclei, at a concentration of 12 g/L, which was added immediately after the introduction of the second portion of ammonium hydrate. Citric acid exhibits weak surface active properties; its CH chain is small, but it is optimal from the point of view of pharmacological use. A fraction of large particles formed in solution was separated by centrifugation in three series of 5 min, with a rotation speed of 800 rpm and 2000× *g*, with intermediate filtration. Trisodium citrate (Na_3_C_6_H_5_O_7_) was used as a stabilizer of the colloidal solution structure at a concentration of 30 g/L.

### 2.2. Absorption Spectra

The absorption spectra were measured with a Cintra 4040 (GBC Cintra 4040, Perth, Australia). The absorption spectra were measured in the quartz cuvettes with an optical path length of 10 mm. The concentration of HEWL was 0.4 mg/mL. Measurements of absorption spectra were carried out in three to six samples for each group. The measurements were carried out at room temperature (~22 °C).

### 2.3. Enzyme Assays of HEWL

The activity of HEWL was examined using the lysis of *M. lysodeikticus* cells at room temperature, as described in [21]. Here, 4 µL of HEWL (5 mg/mL) was collected from the solution with IONPs, diluted 125 times in water or PBS, and added, at 100 µL and a concentration of 40 µg/mL, to 2.5 mL of the micrococcus diluted in 20 mM of K_2_HPO_4_ (pH = 7.0) to an OD of about 0.7–0.8 (λ = 450 nm). The activity was measured by decreases in OD at the same wavelength with a spectrometer (GBC Cintra 4040, Australia) for the first two minutes after the addition of lysozyme. Measurements of lysozyme activity were carried out in three samples for each group. The measurements were carried out at room temperature (~22 °C).

### 2.4. Multi-Angle Dynamic Light Scattering

Zetasizer ULTRA Red Label (Malvern Panalytical Ltd., Malvern, UK) was used to obtain information about particles’ hydrodynamic diameters. Each experiment was done in a plastic cell at 25 °C. A total of 1 mL of each sample was used for experiments. The MADLS method with scattering at the three angles of 174.7°, 90°, and 12.78° was used to measure the size of the molecules and their aggregates. Intensity distributions of hydrodynamic diameters were calculated with ZS Xplorer software. Three independent experiments were made with each solution (5 measurements in each experiment). The concentration of proteins in each solution was 0.4 or 5 mg/mL. The measurements were carried out at 25 °C. For some measurements, the first peak for hydrodynamic diameters of NPs was averaged. This was done as follows: in each data set, the first intensity peak and the hydrodynamic dimensions corresponding to them were found. Then, the values of hydrodynamic diameters for the first intensity maxima were averaged.

Figure 1 shows the hydrodynamic diameters of molecules of protein and NPs, as well as molecules of sodium citrate. The obtained sizes do not change in a wide range of concentrations (see Appendix A). The obtained nanoparticle solution was stable for a year, that is, NPs did not precipitate. The stability of the NP solution was also monitored using MADLS. The first peak of MADLS ~10 nm was reproduced for one year. The concentration of 10 nm nanoparticles also was measured by MADLS (see the results in the Appendix A). We used a solution of NPs with an initial particle concentration of ~10^16^. Solutions with different concentrations of NPs presented in the figures and table were obtained from the initial solution by dilution.

### 2.5. ζ-Potential Measurement

ζ-potential measurements were obtained by Malvern Zetasizer Ultra (Malvern Panalytical Ltd., Malvern, UK) at 25 °C, using ZS Xplorer software. All measurements were made using the automatic attenuation and automatic measurement process (range of runs for each measurement from 10 to 100). The pause between repeats was 60 s. The equilibration time was 120 s.

### 2.6. Fluorescence Spectroscopy

The fluorescence of HEWL in water was studied on a Jasco FP-8300 (JASCO Applied Sciences, Victoria, BC, Canada) spectrometer. The measurements were carried out in a quartz cuvette with an optical path length of 10 mm. All parameters were selected in a way so the peaks for HEWL correspond to approximately 30% of the instrument’s range. Each sample was measured three times. 3D fluorescence figures shows typical spectra. The measurements were carried out at room temperature (~22 °C). The normalization concentration dependence according to fluorescent data was constructed as follows. All samples, both control and with nanoparticles, were excited at a wavelength of 300 nm, and the emission spectra were recorded. The maxima for controls of different concentrations were approximately at the emission wavelength of 337 nm (Table 1). The maxima for a solution of proteins with NPs were in the region of 336–337, and at high concentrations of NPs, they shifted to the region above 340 nm (Table 1) in the emission spectrum. Next, we found the maximum intensity for each NP concentration and normalized it to the intensity of the control maximum.

### 2.7. TEM-Image Microscopy and Energy Dispersive X-ray Spectroscopy

Transmission electron microscopy was done using a Libra 200 Fe HR (Carl Zeiss AG, Oberkochen, Germany). Samples were applied to a gold grid with a titanium holder. Energy dispersive X-ray spectroscopy (EDS) was performed using EDS OXFORD X-Max 100 TLE (Oxford Instruments, Abingdon, UK). All samples for NPs and NPs with protein solution were prepared for TEM as follows. Approximately 0.25 μL drop of solution was applied to a round gold mesh ~4 mm in diameter. The samples were dried at room temperature for 10 min and evacuated.

### 2.8. Image Data Analysis

Images were analyzed with special software to obtain numerical data: characteristic size and the number of particles. The main algorithm of the software is a sequence of the following steps: (1)Raster image is converted into a brightness matrix A (x,y), where x and y are point coordinates for brightness A. Regions with higher A values correspond to the nanoparticle.(2)Threshold T is set, exceeding which means that the pixel belongs to the nanoparticle area. The threshold value was chosen according to the brightness histogram of the original image. We form the matrix *I*(x,y): $I\left(x,y\right)=\{\begin{array}{c} 0,\;\mathrm{if}\;A(x,y)<T\\ 1,\;\;\mathrm{if}\;A(x,y)\geq T \end{array}$(3)We form the matrix of the number of neighbors *b*(x,y): $b\left(x,y\right)=\sum_{i=-1}^{1}\sum_{j=-1}^{1}I(x+i;y+j)-I(x,y)$(4)We form a matrix of boundary pixels: $B\left(x,y\right)=\{\begin{array}{c} 0,\;\mathrm{if}\;b(x,y)<3\\ 1,\;\mathrm{if}\;3\leq b(x,y)\leq 5\\ 0,\;\mathrm{if}\;b(x,y)>5 \end{array}$(5)We form a matrix of “internal” pixels: $F\left(x,y\right)=\{\begin{array}{c} 0,\;\mathrm{if}\;b(x,y)<6\\ 1,\;\mathrm{if}\;b(x,y)\geq 6 \end{array}$

Next, for each boundary pixel, we construct segments connecting it with all other boundary pixels. For each such segment, we calculate its length using the Pythagorean Theorem and calculate the percentage of the passage of the segment through the “internal” pixels. If this percentage is greater than 90%, we include the length of the segment in the set to build the size distribution. The peak value of segment length distribution histograms was taken as the probable size of nanoparticles obtained from the analysis of TEM images.

### 2.9. Fibroblast Isolation

The isolation of fibroblasts from murine lungs was performed according to the standard protocol. Additional details about mice and cells isolation can be found in the Appendix A and Ref. [22].

### 2.10. Fluorescence Microscopy

Cell viability, cell surface area, and ROS production were evaluated with fluorescence microscopy. Cells were incubated for 24 h with corresponding concentrations of NPs, citrate, lysozyme, and their combinations. Hoechst 33342 and propidium iodide (PI) were used to evaluate cell morbidity [23]. Fluorescent microscopy with staining by Hoechst 33342 and propidium iodide is a widely used cell viability standard test (including NP cytotoxicity assays and drug screening) [24,25,26,27,28]. Rhodamine-123 was used to contrast cytoplasm and evaluate mitochondrial membrane potential level [29]. Immediately after incubation, a coverslip with cells was placed in a coverslip chamber (RC-40LP, Warner Instruments LLS, Hamden, CT, USA), washed with PBS, and stained with 5 µg/mL Hoechst 33342 and 5 µg/mL rhodamine-123 (both ThermoFisher, Waltham, MA, USA) for 30 min at 37 ℃. Then, the sample was washed with PBS and stained with 2 μM PI (ThermoFisher, Waltham, MA, USA) for 1 min. The samples were analyzed using a fluorescent microscope DMI4000 B (Leica Camera AG, Wetzlar, Germany) equipped with an SDU-285 digital camera (SpetsTeleTekhnika, Moscow, Russia). Fluorescence spectra were recorded at the following excitation/emission wavelengths: 350/470 for Hoechst 33342 (filter cube D, Leica, Germany), 488/520 (filer cube I3, Leica Camera AG, Wetzlar Germany) for Rhodamine-123, and 540/590 for PI (filter cube TRITC Leica Camera AG, Wetzlar, Germany). Light emitting diodes (LEDs) M375D2m and M490D3 (Thorlabs, Newton, NJ, USA) and a white LED (Cree Inc., Durham, NC, USA) were used as light sources for excitation of Hoechst 33342, Rhodamine-123, and PI fluorescence, respectively. All images were taken at the same LED current of 100 mA for M375D2m (Hoechst 33342), 250 mA for M490D3 (Rhodamine-123), and mA for the white LED (PI). Exposition time was the same in all experiments: 500 ms for Hoechst 33342, and 1000 ms for Rhodamine-123 or PI. The obtained coefficient of the detector was equal to x523 and was the same for all fluorophores and experimental conditions.

WinFluorXE software (version 3.8.7, J. Dempster, Strathclyde Electrophysiology Software, University of Strathclyde, Glasgow, UK) was used for data acquisition. Data were collected as 12-bit grayscale images. Subsequent analysis was performed using the ImageJ version 2.0.0-rc-69/1.52p (Java 1.8.0_171) software (NIH, Bethesda, MD, USA). For each experimental variant, at least five samples were analyzed. At least 200 cells were analyzed in each sample.

Regions of interest (ROIs) were determined using automated standard ImageJ procedures “Threshold” and “Analyze particles”. Parameters for analysis and ROI determination were selected as a result of preliminary experiments. For images, 1392 × 1024 pixels obtained with magnitude ×20 were used with the following parameters: “size” = 100–750 and “circularity” = 0.10–1.00. Nuclei had various Hoechst 33342 and PI fluorescence intensities. Therefore to ensure that all nuclei entered the analysis, we carried out a series of “Threshold” and “Analyze particles” procedures on each image. Images were converted to 8-bit before ROI determination. Threshold levels were from 5 to 255 a.u. with an increment of 5 a.u.; ROI was saved as binary masks. Further, all ROIs were combined, with duplicates deleted. All procedures were integrated into automated macros.

Fluorescent intensities in ROIs were measured by standard procedure «mean/average gray value». This procedure calculates the sum of the gray values of all pixels in the ROI divided by the number of pixels. The background was subtracted from each image previous to the fluorescence intensity measurement. Each image was duplicated and transformed with a medial filter with 100 × 100 pixels. The obtained picture was subtracted from the corresponding initial picture by the standard procedure “Image calculator”. 

### 2.11. Apoptosis Assay

The apoptosis assay was performed with AlexaFluor488-conjugated Annexin V (AF488-Annexin V) [30]. Cells were stained with 5 μg/mL Hoechst 33342 for 30 min and washed with binding buffer (50 mM HEPES, 700 mM NaCl, 12.5 mM CaCl_2_, pH 7.4, all from Sigma-Aldrich, St. Louis, MO, USA) three times. Washed samples were stained with AF488-Annexin V (diluted 1:100) for 30 min, washed with binding buffer, and stained with 2 μM PI for 5 min. The samples were analyzed using a fluorescent microscope DMI4000 B (Leica Camera AG, Wetzlar, Germany) equipped with an SDU-285 digital camera (SpetsTeleTekhnika, Moscow, Russia). Fluorescence spectra were recorded at the following excitation/emission wavelengths: 350/460 for Hoechst 33342, 488/520 for AlexaFluor488, and 540/590 for PI. Fluorescence images were recorded at the following excitation/emission wavelengths: 350/470 for Hoechst 33342 (filter cube D, Leica Camera AG, Wetzlar, Germany), 488/520 (filer cube I3, Leica Camera AG, Wetzlar, Germany) for AlexaFluor488, and 540/590 for PI (filter cube TRITC Leica Camera AG, Wetzlar, Germany). WinFluorXE software (version 3.8.7, J. Dempster, Strathclyde Electrophysiology Software, University of Strathclyde, Glasgow, UK) was used for data acquisition. Data were collected as 12-bit grayscale images. Subsequent analysis was performed using the ImageJ version 2.0.0-rc-69/1.52p (Java 1.8.0_171) software (NIH, Bethesda, MD, USA). AF488-Annexin V+ cells were indicated as apoptotic. PI+ cells were indicated as non-viable. The ratio of the amount of AF488-Annexin V+PI+ cells to all dead PI+ cells was calculated to evaluate the percentage of the cell that died via apoptosis. For each experimental variant, at least five samples were analyzed. At least 200 cells were analyzed in each sample.

## 3. Results

A TEM image of TSC-IONPs particles is shown in Figure 2a,b. Images obtained with a transmission microscope suggest that synthesized TSC-IONPs have a spherical shape with a ~3 nm size, and they form big and small clusters. There is a possibility that NP clusterization in the figures may occur as a result of sample preparation for TEM. However the TEM data are in good agreement with the Multi-angle Dynamic Light Scattering (MADLS) data, which were obtained in solution (Figure 1). When nanoparticles interact with hen egg-white lysozyme (HEWL) molecules, the latter precipitate. In this case, the clusters of nanoparticles break up, and TSC-IONPs are almost evenly distributed over the protein aggregates (Figure 2c). After the addition of lysozyme, no individual NP clusters were seen in the samples (see Appendix A). Figure 2c shows the distribution of segments between all points of the boundaries of the NPs. The maximum in the distribution of segments corresponds to the most probable size of NPs (see Section 2). It can be seen from the distribution that the most probable size of the NPs before interaction with the protein (Figure 2b and green line in Figure 2d) lies in the region of 6.5–8 nm. When interacting with lysozyme (Figure 2c and red line in Figure 2d), the most probable size of the NPs becomes 2.8 nm. This peak corresponds to individual NPs formed after the decay of the initial aggregates of NPs.

In addition, energy dispersive X-ray spectroscopy (EDS) analysis of a sample TSC-IONPs with HEWL was performed (Figure 3a). EDS analysis shows that iron particles in the sample are not distributed in the form of initial agglomerates. On the other hand, the iron molecules are less evenly distributed compared to the C and O molecules, whose uniform contribution is due to proteins. The results of energy dispersive spectroscopy show that the expected elements of the NPs themselves (Fe, O) and trisodium citrate (Na, C, O) and proteins (C, O) are present in the samples (Figure 3b). In addition, there are pads for TEM imaging (Au, Ti). Chlorine is most likely present in the sample due to the iron chlorides used in the preparation of the NPs.

The formation of large micron protein aggregates upon the addition of sufficiently high concentrations of TSC-IONPs is confirmed both with a fluorescent microscope (Figure 4a) and with absorption on a spectrophotometer (Figure 4b). Note that when NPs are added at a concentration of 10^13^ mL^−1^, the absorption peak shifts from 280 nm to longer wavelengths. It was decided to check how much the shell NP material can affect absorption in the same region. The protein absorption didn’t change, even with 1 M TSC (Figure 4b). TSC is used in medicine as an anticoagulant, and it is known that protein precipitation occurs when adding 500 mM TSC [31].

Interesting results were also obtained for measurements of the fluorescence of proteins with TSC-IONPs and TSC (Figure 5). Protein fluorescence decreased upon interaction with NPs, for a protein solution with both 5 mg/mL and 0.4 mg/mL (Figure 5a). The emission of HEWL samples of various concentrations upon the addition of NPs in various concentrations is shown in Figure 5b–f. The samples were excited at 300 nm, the emission peak for different concentrations of lysozyme in the control was ~337 nm, and when TSC-IONPs were added, it shifted to the long wavelength region (Figure 5c–f, Table 1). For comparison, Figure 5b shows normalized data, where the intensities of fluorescence maxima for each concentration of NPs and HEWL were normalized to the intensities of the corresponding control. The same fluorescent concentration curves for proteins at 0.4, 5, and 100 mg/mL may indicate that the decrease in fluorescence is primarily associated with the addition of NPs, which absorb well in this range (see the top inset in Figure 4b). But the change in the course of the curve at a protein concentration of 0.01 mg/mL indicates the interaction of protein NPs. Figure 5b–f shows that the decrease in fluorescence caused by protein and NPs aggregation depends on the HEWL concentration, and aggregation begins at a lower NP concentration with low concentrations of protein (0.01 mg/mL). The aggregation occurred in the same way for high protein concentrations (0.4–100 mg/mL, Figure 5b). In contrast to the interaction of NPs, the interaction with TSC 1 M depends to a greater extent on the protein concentration. Minimal but noticeable changes in fluorescence were for protein concentrations of 0.4 mg/mL, while fluorescence for protein concentrations of 5 mg/mL decreased three-fold (Figure 5a). 

**Figure 5 nanomaterials-12-03960-f005:**
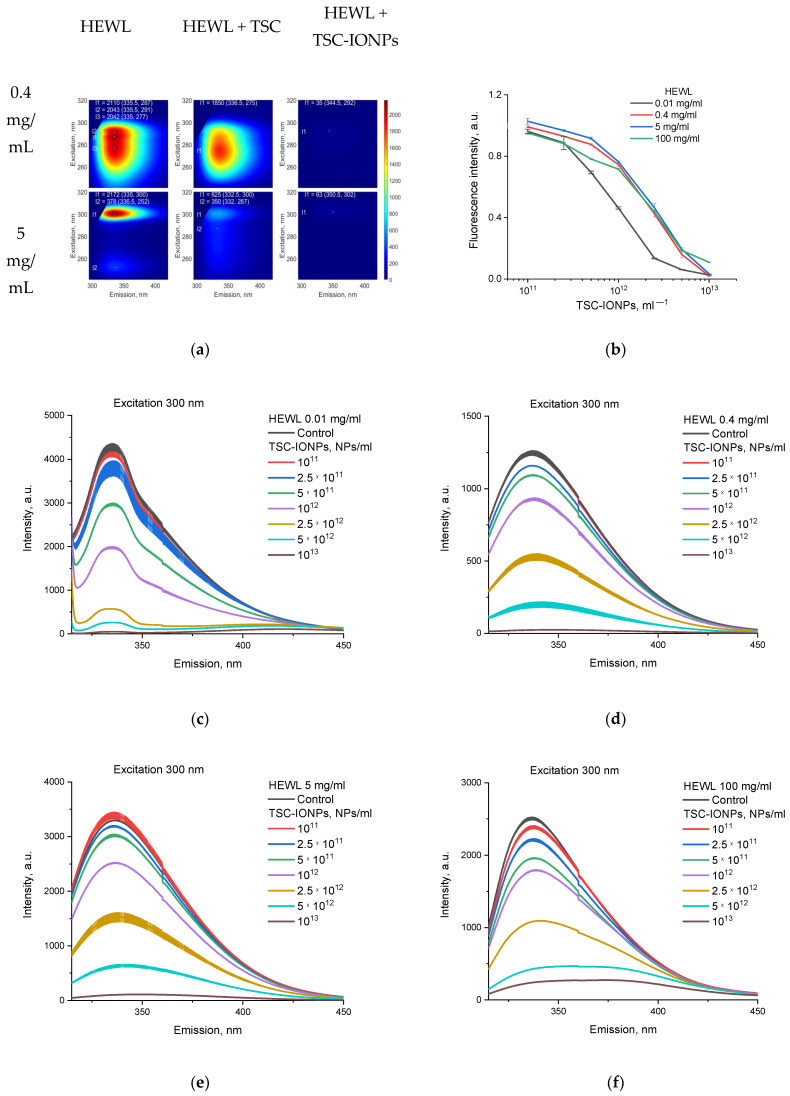
(**a**) Fluorescence for HEWL (0.4 and 5 mg/mL), HEWL with TSC (1 M), and HEWL with TSC-IONPs (10^13^ NPs/mL). The points show the positions of the maxima. The intensities and coordinates of the maxima are written at the top of the graph. (**b**) Normalized emission of HEWL samples of various concentrations upon the addition of TSC-IONPs. (**c**–**f**) Emission of 0.01, 0.4, 5, and 100 mg/mL HEWL samples with different concentrations upon the addition of TSC-IONPs. Results are presented as mean and SD for three independent experiments.

**Table 1 nanomaterials-12-03960-t001:** Fluorescence peak position (mean and standard deviation) when excited on 300 HEWL samples at 0.01, 0.4, 5, and 100 mg/mL. See Figure 5.

HEWL with TSC-IONPs mL^−1^	0.01 mg/mL	0.4 mg/mL	5 mg/mL	100 mg/mL
0	335.2 ± 0.1	336.9 ± 0.3	336.3 ± 0.2	336.9 ± 0.4
10^11^	335.3 ± 0.1	336.7 ± 0.3	335.7 ± 0.1	337.7 ± 0.2
2.5 × 10^11^	335.7 ± 0.1	336.9 ± 0.6	335.9 ± 0.7	337.4 ± 0.5
5 × 10^11^	335.5 ± 0.1	337.1 ± 0.5	336.5 ± 0.6	338.1 ± 0.5
10^12^	335.1 ± 0.1	337.7 ± 0.3	337.0 ± 0.4	339.1 ± 0.3
2.5 × 10^12^	333.4 ± 0.4	339.5 ± 0.5	338.5 ± 0.1	340.4 ± 0.2
5 × 10^12^	335.6 ± 0.3	341.9 ± 1.1	342.1 ± 0.9	358.6 ± 1.4
10^13^	-	346.8 ± 3.1	349.3 ± 1.5	374.7 ± 2.2

In addition, we observed a shift in the fluorescence maximum upon the addition of NPs (Figure 5c–f, Table 1). Characteristically, this shift depends on the protein concentration (Table 1). This indicates that the protein interacts with nanoparticles to form aggregates.

Data on the activity of HEWL (see Appendix A) indicate that when interacting with TSC-IONPs in a wide range of NP concentrations (from 10^9^ to 10^13^ mL^−1^), the protein activity does not change compared to the control and remains in the region of 43 ± 7 U/mg. This means that the protein either does not significantly change its native conformation upon formation of aggregates with NPs or quickly returns to it when diluted more than 100-fold in the test with lysis of *M. lysodeikticus* cells. This dilution occurs during the activity test, with the initial protein concentration of 5 mg/mL reduced to 40 microns/mL.

Figure 6a–c shows the particle diameter distributions weighted by intensities, obtained by MADLS for HEWL (0.01, 0.4, and 5 mg/mL) and TSC-IONP (from 10^11^ to 5 × 10^13^ mL^−1^) solutions. No concentration dependence of NPs sizes was found. In the studied concentration range from 10^11^ to 5 × 10^13^, the peaks of the distribution intensities were the same as for the concentration of 10^13^ in Figure 1. It can be seen from all the graphs that with an increase in the concentration of NPs, the size of aggregates of the protein with NPs increases. According to Figure 6a, intensity peaks are not visible in the region of 1 nm associated with individual HEWL molecules. This is explained by the fact that the region is out of the sensitivity range of the device. The device manufacturer recommends 0.1 mg/mL protein as a minimum concentration for valid measurement. At a protein concentration of 0.4 mg/mL, the intensity peaks in the region of 1 nm are also remarkably smaller than at 5 mg/mL. In concentration dependencies with high protein content, single peaks of HEWL molecules disappear when TSC-IONPs are added at a concentration of ~10^12^–2.5 × 10^12^ (Figure 6b,c).

Figure 6d shows the dependences of the average particle sizes in solution for different concentrations of NPs (from 10^11^ to 5 × 10^13^ mL^−1^) and different concentrations of HEWL (0.01, 0.4, and 5 mg/mL). The average particle/aggregate size is plotted depending on the concentration of NPs and proteins. The conversion from the distribution of intensities to the distribution of the number of particles was performed using the ZS Xplorer software (Malvern Panalytical Ltd., Malvern, UK). To find the average hydrodynamic diameter from the distribution data, the following formula was used: D=∑i=1nIidin, where $I_{i}$—the percentage of the total number of particles, $d_{i}$—hydrodynamic diameter, *i* —the sampling step, *n*—the number of points in the distribution (maximum 150 for MADLS).

As can be seen from the data presented in Figure 6d, for the first two concentrations (10^11^ and 2.5 × 10^11^ mL^−1^) of nanoparticles for 0.01 mg/mL HEWL solution, the average hydrodynamic diameters were greater for the concentration of protein of 0.4 or 5 mg/mL. However, the Zetasizer does not allow registration of the protein at 0.01 mg/mL. Thus, nanoparticles and protein aggregates associated with NPs are registered in the solution. With the increase of the concentration of nanoparticles, the formation of aggregates for a given solution occurs at lower concentrations of NPs, which is consistent with the fluorescence data (Figure 5b). The decrease in the average size of aggregates for 0.4 mg/mL of HEWL and the concentration of 2.5 × 10^13^–5 × 10^13^ NPs/mL is most likely caused by their passage through a point where the ζ-potential is close to zero (Table 2). The rate of appearance of such aggregates (Figure 6) is quite fast. Figure 7a shows the time dependence of the appearance of protein aggregates with NPs. It is worth adding 3–4 min to point 0 on the time dependence, during which the MADLS measurement takes place. It can be seen that the size of the aggregates reaches a plateau after 5–10 min and then practically does not change. The observed aggregation is not irreversible. We were able to detect single lysozyme molecules after 10 s shaking on a Biosan V1-plus vortex (Figure 7b).

Table 2 shows the values of the potentials, pH, and hydrodynamic diameter of the first peak taken by the MADLS method. The value of the ζ-potential −32.1 ± 2.3 mV and size 10.9 ± 0.6 nm TSC-IONPs turned out to be similar to values obtained on similar particles by other authors [32]. It can be seen from the table that when TSC-IONPs are added to HEWL, the ζ-potential of the particles and pH in the solution changes. Initially, positively charged proteins, when interacting with negatively charged TSC-IONPs, lose their charge during the formation of aggregates (Figure 6). At the same time, the pH of the solution also changed. It may seem that the formation of aggregates is due to a change in pH from 4 to 8. However, the aggregation was not observed by the MADLS method even with a change in pH up to 12 by using NaOH.

To test the toxicity of nanoparticles, a series of experiments were carried out with 24-h seeding of mouse fibroblast cells. Figure 8 shows graphs of cell viability, area per cell, and rhodamine intensity, depending on the concentration of TSC-IONPs. All parameters were determined from the fluorescent microscope images with various dyes (Figure 9). Cell viability was determined by the number of PI-stained death cell nuclei and the number of Hoechst-stained live cell nuclei. Cell area was determined by the dye rhodamine 123, which binds active mitochondria [33]. Fluorescent microscopy with staining by Hoechst 33342 and propidium iodide is a widely used cell viability standard test (including NP cytotoxicity assays and drug screening) [24,34]. This method has reproducibility comparable to the MTT test [22,35].

It turned out that in a wide range of concentrations up to 5 × 10^12^ mL^−1^, NPs do not have a significant effect on cell viability (Figure 8 and Figure 9). Further, the viability sharply decreases when TSC-IONPs are added. At 10^13^ NPs concentrations, only a small percentage of cells remains viable (2.1 ± 1.3%). For comparison, iron concentrations are shown below in Figure 8. The concentrations were calculated according to MADLS data (Figure 1) from the number of 10 nm NP aggregates, each of which contains approximately 15 NPs 2.8 nm in diameter (Figure 2b).

The area of cells increases (grows ~57% compared to control, statistically significant difference *p* < 0.01, *t*-test) up to the concentration of NPs 5 × 10^11^, and then decreases relative to the control after TSC-IONP concentrations 7.5 × 10^12^ (Figure 8). One-way ANOVA showed statistical significance F (7, 111) = 9.86 with *p* << 0.001 in differences of the mean of the group. This indicates that at least one of the groups does not lie in the total group. Statistically significant changes (one-way ANOVA F (7, 94) = 5.1505, *p* << 0.001) were also observed for the rhodamine intensity when adding TSC-IONPs (Figure 8). Although the picture is not as clear as with the area and there is large SD, there is a tendency to increase the rhodamine intensity with an increase in the concentration of NPs. The same data for TSC is included in the Appendix A, because no significant differences were found between the control and cells treated with 0.1 mM TSC.

As mentioned above, an increase in the fluorescence intensity of Rhodamine 123 is associated with a decrease in the mitochondrial membrane potential [36]. It is believed that a decrease in the membrane potential is associated with disruption of the functioning of mitochondria and the onset of processes from cell destruction to mitophagy [37]. In addition, a decrease in the mitochondrial membrane potential, including the one determined by the fluorescence of Rhodamine 123, can be a marker of the early stages of apoptosis [38], which is consistent with data on cell death and the decrease in the area at high concentrations of NPs (Figure 8 and Figure 9). Mean fluorescence intensity can depend on the area of a cell. But in our case, differences in Rhodamin-123 fluorescence intensities between the control and IONP-treated cells cannot be explained only by a change of cell area. Cell area variation is less than fluorescence intensity value variation; therefore, there are additional reasons for fluorescence variation. Such a sharp sigmoid curve for cell viability is somewhat unexpected. Toxic effects previously observed by other authors showed a tenfold decrease in survival (from ~90% to ~10%) with a change in the concentration of iron (for IONPs ~40 nm) in the solution by two orders of magnitude from 0.1 mM to 10 mM [39]. However it should not be excluded that such a sharp sigmoidal curve is a feature of the cell line.

Using the AF488-Annexin V stain, we examined the percentage of apoptotic cells among all dead cells (Figure 10). The figure shows that the percentage of apoptotic cells increases with an increase in the concentration of NPs and reaches three-quarters at a concentration of 10^13^ mL^−1^.

In separate survival experiments, HEWL (0.4 mg/mL, final concentration in cell solution) or TSC (50 mM, final concentration in cell solution) was also added to the TSC-IONP (10^13^ mL^−1^) solution before adding NPs to the cell culture (see Appendix A). The aim was to check how much the toxicity of the particles would decrease if they were bound to a protein or additional citrate. In other studies, with additional protein coating of NPs, the toxicity of NPs is usually greatly reduced [40]. However, no effect on cell viability was observed. Cell viability was 1.1 ± 0.8% with the addition of NPs and HEWL (0.4 mg/mL) and 3.4 ± 3.1% with the addition of NPs and TSC (50 mM). In addition, only TSC-IONP (10^13^ mL^−1^) cell viability was 2.0 ± 0.6%.

## 4. Discussion

One of the difficulties in using magnetic NPs in practice [41,42] is their tendency toward clusterization, which is accompanied by a significant change in their properties. The aggregation of magnetic NPs is a consequence of the action of van der Waals and Coulomb forces and the forces of magnetic dipole interaction. Van der Waals forces are short-range; they decrease in proportion to approximately the sixth power of the distance between the particles, while the energy of the Coulomb and dipole interactions decreases much more slowly. To eliminate clusterization, the NPs are coated with a surfactant or neutral polymer. This imparts a surface electric charge to nanoparticles or forms a shell that prevents particles from approaching [43]. In the latter case, the van der Waals forces can be neglected. In addition, clusterization is affected by the viscosity of the liquid medium, thermal fluctuations, and the magnetic anisotropy of the particle matter. All these factors lead to a complex dependence of the interaction energy of NPs with their characteristics, coatings, and the hydrodynamic properties of the suspension.

The TSC-IONP particles prepared herein were quite small, at ~3 nm in diameter. The small particle size has its advantages and disadvantages. One of the advantages is that nanoparticles form stable clusters ~10 and ~ 50 nm in size. Clusters are visible on the MADLS (Figure 1) and remain visible even after vacuumization for TEM (Figure 2a,b). For example, in the works with NPs presented in the Section 2, the size of NPs ranges from 10 to 50 nm, but in TEM pictures, they form large micron-sized clusters.

An advantage of the formation of stable clusters is the decrease in the ζ-potential of the particles. Typically, NPs in solution are considered stable if the absolute value of the ζ-potential is greater than 30 mV [44]. However, such nanoparticles are not suitable for medical purposes. Due to the strong charge on the surface, they form aggregates with proteins with opposite charges in tissues and blood plasma [45,46]. In the case of particles used in the research, the ζ-potential of new particles changes from −40 mV to −30 mV within a few days and stabilizes due to the formation of clusters.

The fabrication of coated NPs is complicated by the control of the coating condition. Shells or coatings consisting of atoms lighter than metal NPs are usually not visible on TEM images (Figure 2b). Qualitatively, the presence of shells can be shown using EDS analysis. In our case, the peak of the Na element is visible on the spectrum (Figure 3b and Appendix A). However, using EDS analysis it is very difficult to show that an element is on the shell and not in the solution. In particular, on the spectra, chlorine can be seen, which remains from the original reactants in the solution (Figure 3b). It was not possible to determine the distribution of elements for sodium (Figure 3a), since this method requires long-term measurements, and Na “burns out” under the beam. The integrity of the shell can only be measured indirectly, with the precipitation of NPs and their toxicity, which sharply increases for uncoated particles.

The number of TSC molecules per IONP can be counted. Based on the TEM image, the nanoparticle size was taken to be 2.8 nm, and the area of the NPs was ~35 nm^2^. The literature contains data on the density of citrate molecules on the surface of metal nanoparticles. It has been estimated both experimentally and theoretically, but mainly for gold NPs, and it varies from 0.4 to 4 molecules per nm^2^ of the NP surface [47]. Based on these estimates, the number of citrate molecules per nanoparticle ranges from 14 to 140 or from 0.2 to 2.3 nM for 10^13^ mL^−1^ NPs. Initially, 30 g/L of TSC was used to stabilize IONPs in solution that was diluted 10^3^ times. Therefore, in the experiments with 10^13^ TSC-IONPs, the solution contained about 0.1 mM TSC. Lower concentrations (by 15–20 times) of TSC in the manufacture of TSC-IONPs resulted in greater toxicity. This indicates that sodium citrate molecules are not rigidly fixed on the NP surface and can move away from the surface and be replaced by new ones from the solution. It should be added that TSC 0.1 mM did not lead to changes in cell viability. Cell viability at 24 h was 98.7 ± 1.0% at TSC 0.1 mM compared with the control: 99.7 ± 0.4%. TSC at a concentration of 0.1 mM did not lead to protein aggregation (first peak MADLS = 1.15 + 0.03 nm, ζ-potential = 28 ± 3 mV) compared to protein at a concentration of 5 mg/mL (first peak MADLS = 1.1 + 0.1 nm, ζ-potential = 27 ± 5 mV, Table 1). Even by increasing the concentration of TSC to 1 M, protein aggregation occurred weakly and only at high protein concentrations (Figure 4 and Figure 5).

It is interesting to compare the properties of uncoated IONPs and IONPs coated with sodium citrate. Unfortunately, our uncoated particles were unstable and precipitated. However, in the range of published research, it has been shown that it is possible to stabilize uncoated NPs and measure both particle properties using TEM and MADLS and toxicity. In the Ref. [48], the cytotoxicity of some tests of coated nanoparticles was higher than that of NPs without shells (for 10–12 nm IONPs). Uncoated 10 nm particles (TEM measurement) formed 180 nm aggregates when measured in DLS. Trisodium citrate-coated 12 nm particles formed aggregates at 320 nm [48]. The presented example is not the only one. For instance, in [15] the hydrodynamic radius of IONPs measured using DLS significantly exceeded the sizes measured using TEM and depended on the solvent.

In addition to the very fact of aggregation in solution with NPs, the question arises of how accurately DLS measures the sizes of molecules or aggregates. For example, a sufficiently large HEWL molecule shows a smaller particle size than TSC (Figure 1). This result seems doubtful. Most likely the TSC peak is caused by contaminants; indeed, the fluorescence spectrum shows a small but noticeable peak at 316 nm excitation and 410 nm emission (see Appendix A). On the other hand, when measured by the DLS method, it is in water that single protein molecules give sizes smaller than, for example, those obtained using X-ray diffraction analysis [49]. The hydrodynamic diameter of lysozyme measured using DLS under control conditions was ~1.2 nm (Figure 1). This is an interesting phenomenon that many researchers working with the DLS method may have encountered. Measurement of DLS using the hydrodynamic diameter of macromolecules is strongly dependent on the solvent and must take into account corrections for viscosity and the refractive index necessary for fine particle sizes, using the autocorrelation function. For example, the lysozyme we measured earlier in 50 mM Tris-HCl (pH = 8.0) had a size of 3.8 nm [50]. In this work, the hydrodynamic diameter of lysozyme increased with the addition of TSC and reached 2.5 nm for 1 mM TSC solution (Appendix A). Other authors have also observed particle sizes in DLS measurements for proteins [51] and even for carboxylated latex beads [44]. Moreover, in the latter case, the differences in particle sizes between different solvents reached one order of magnitude. Undoubtedly, DLS data should be treated with caution and considered together with TEM data. On the other hand, TEM microscopy is also not ideal, since the sample preparation method involves moving from an aqueous medium to a substrate and a vacuum.

Thus, in this work, it is shown that trisodium citrate-coated iron oxide nanoparticles of ~3 nm form stable ~10 nm clusters. When interacting with lysozyme, the clusters disintegrate, and the formation of submicron aggregates with protein occurs. At the same concentration of NPs, cellular effects were studied. Before moving on to cellular effects, we aimed to assess how the magnetic field generated by TSC-IONPs can be toxic to cells.

IONPs are known to create around them a magnetic field that is much stronger than the geomagnetic field of about 0.05 mT. Inside a layer with a thickness of the order of the particle radius, the magnetic field reaches tens of mT [52]. These magnetic fields can change the rate of biochemical reactions involving spin-correlated pairs of radicals. For example, the electron transport that controls the activity of cryptochromes occurs with the formation of radical pairs and depends on the magnetic field. An analysis of the possible toxicity of IONP clusters would be incomplete without an assessment of their magnetic interaction. In addition, such a magnetic interaction is one of the factors of nanoparticle aggregation [4,5], which is accompanied by a significant loss of their useful properties.

To avoid clusterization of NPs, production technologies provide their coating with a surfactant or a neutral polymer. This imparts an electric charge to NPs or forms a shell that prevents particles from approaching [43]. In addition, clusterization is affected by the viscosity of the liquid medium, thermal fluctuations, and the magnetic anisotropy of the particle matter. All these factors lead to a complex dependence of the interaction energy of NPs on their characteristics, coatings, and the hydrodynamic properties of the suspension.

In estimates, magnetic anisotropy is often neglected, since thermal perturbations are accompanied by the random orientation of particles. It is also assumed that the suspension is unstable if thermal perturbations destroy the resulting particle complexes: U/kT≪1, where U is the maximum absolute value of the potential energy of interaction between a pair of particles. The NPs used in this work, having an approximately spherical shape and a core radius *r* of about 1.5–2 nm, are superparamagnetic. The direction of their magnetic moment fluctuates at microwave frequencies and higher. This means that the direction of the moment in relatively slow aggregation processes obeys Boltzmann statistics. In approximate calculations, NPs are usually considered single-domain, i.e., their magnetic moment m is equal to vJ, where $v=4\pi r^{3}/3$ is the particle volume, and J=480 G (CGS units) is the saturation magnetization of magnetite Fe_3_O_4_. The IONPs used are coated with a citrate or protein shell, the thickness of which is denoted by d. Neglecting the surface charge of the shell, the minimum potential energy corresponds to the distance between the centers of the particles s=2(r+d) and is determined by the magnetic dipole interaction. For calculation, the thickness of the citrate shell has been taken to be about 1 nm; the thickness of the protein shell has been assumed to be equal to the hydrodynamic diameter of the lysozyme molecule [53], i.e., 4 nm.

The magnetic contribution *U* to the energy of interaction between two IONPs was estimated, for example, in Ref. [54]. The authors used Boltzmann statistics, and their result is reduced to the following relations (CGS units):(1)$$\frac{U}{kT}=\frac{\sum_{n=1}^{\infty }2nK_{n}x^{2n}}{\sum_{n=0}^{\infty }K_{n}x^{2n}},\;K_{n}=\frac{{\left(n!\right)}^{2}}{{\left[\left(2n+1\right)!\right]}^{2}}\sum_{j=0}^{n}\frac{\left(2j\right)!}{{\left(j!\right)}^{2}},\;x=\frac{2m^{2}}{kTs^{3}}$$

The relative value of the magnetic interaction U/kT calculated from Equation (1) is shown in Figure 11. Since the radius of our IONPs is about 1.5–2 nm, their magnetic interaction energy according to Equation (1) and Figure 11 is more than a hundred times lower than kT for both bare and coated particles. The observed clusterization of the studied nanoparticles occurs without the participation of their magnetic interaction. The magnetic field produced by an aggregate even of a few nanoparticles is (a) significantly less than the field of a single particle due to the closure of magnetic fluxes inside the cluster, and (b) is concentrated mainly inside the protein shell, since it thickness significantly exceeds the particle radius. Thus, the magnetic interaction is also irrelevant to the observed toxicity.

As for the observed cytotoxicity of TSC-IONPs, starting from a certain concentration of NPs (~5 × 10^12^), there is a sharp drop in cell viability (Figure 8 and Figure 9). The loss of viability is possibly related to apoptosis or necrosis, although the photographs do not show signs of the destruction of the membranes (Figure 9). The decrease in cell size observed at concentration ~7.5 × 10^12^ confirms this possibility (Figure 8). A decrease in cell size when adding high content of NPs may indicate the induction of apoptosis. A decrease in cell size during apoptosis has been shown in the literature [55]. Another process is also interesting: the growth of cell area with the addition of TSC-IONPs at concentrations 5 × 10^10^–5 × 10^11^ (Figure 8). A change in the shape of cells may indicate a change in the properties of their surface, in particular, changes in the composition or functioning of surface proteins. For negatively charged NPs, the ability to change the protein composition on the surface of eukaryotic cells has been shown [56]. An increase in size can also be associated with an increase in cell adhesion to the substrate, leading to “cell spreading” [57]. The increase in cell area under the action of NPs is consistent with a previously published study [23].

We found a decrease in the size of nuclei during cell death (Figure 9). Usually, a decrease in the size of nuclei (pyknosis) occurs at the early stages of cell death during apoptosis or necrosis [58,59]. Based on a decrease in cell size, it can be assumed that NPs cause cell death through apoptosis, since necrosis is characterized by an increase in cell size due to impaired cell membrane permeability and water “leakage” [60]. This is confirmed by our measurements. With an increase in the concentration of NPs, not only does the number of dead cells increase but also the percentage of apoptotic cells among them increases. At NP concentrations up to 10^13^ mL^−1^, three-quarters of the cells among the dead went into apoptosis (Figure 10). Cell death may occur due to damage to the cell membrane. Cell membrane disruption is a common mechanism of IONP cytotoxicity against both eukaryotic and prokaryotic cells [61,62]. Cell membrane damage can also be one of the causes of cell death in our study. Clarification of the mechanism of cell death caused by NPs is planned in further studies.

In the concentration range of TCS-IONPs 10^12^–10^13^ mL^−1^, the formation of aggregates was observed in experiments with HEWL. The formation of aggregates was recorded by several methods: MADLS (Figure 6), optical absorption (Figure 4b), fluorescence (Figure 5), TEM (Figure 2c), and conventional microscopy (Figure 4a). At the same time, it is characteristic that nanoparticles do not float in solution surrounded by a protein “crown”, but form aggregates with HEWL molecules similar to “buns with raisins” (Figure 2c).

Particularly interesting data were obtained when measuring fluorescence. When TSC-IONPs were added to the protein solution, the fluorescence decreased (Figure 5). The decrease in fluorescence, in our opinion, occurred for two reasons. First, the NPs themselves absorb well in the range of 260–300 nm (see the inset at the top right of Figure 4b). However, this absorption is not enough to explain the dependence of the fluorescence decay on the protein concentration (Figure 5b). For example, for a concentration of NPs of 10^12^ mL^−1^, the decrease in fluorescence for a HEWL concentration of 0.4 mg/mL and above is about 20%, while for a concentration of 0.01 mg/mL it is already 50%. In addition, this does not explain the shift in the fluorescence maximum that is observed when NPs are added (Figure 5, Table 1). Moreover, this shift increases with a greater concentration of lysozyme. Such shifts in fluorescence were previously recorded during denaturation, both thermally [63] and with guanidine hydrochloride [64]. Shifts in fluorescence and the appearance of new peaks in the long-wavelength region were also observed during the formation of amyloid fibrils [65]. The second and main reason, in our opinion, is protein aggregation, as a result of which less excitatory light reaches fluorescent amino acids (Trp, Tyr, Phe), and absorption of emission light (330–360 nm) by the protein aggregates also occurs (Figure 5b–f).

At first glance, the concentration dependences of the fluorescence of TSC-IONPs seem unexpected (Figure 5b). The curves for the concentration of HEWL 0.4–100 mg/mL practically coincide with each other, and the curve with the smallest protein concentration differs from the others. In it, aggregation occurs at a significantly lower concentration of NPs. A halving of fluorescence occurs at a protein concentration of 0.01 mg/mL and a concentration of TSC-IONPs 10^12^ mL^−1^. The ratio of protein molecules to 3 nm NPs in aggregates is approximately 2.7 × 10^4^. Such a large ratio indicates that one particle is capable of aggregating thousands of protein molecules. It is not entirely clear how this is possible given the approximately equal geometric size.

The formation of aggregates according to fluorescence data did not depend on protein concentration in the range of 0.4–100 mg/mL (Figure 5b). An interesting phenomenon of shifting aggregation to low concentrations of TCS-IONPs at lower HEWL concentrations (0.01 mg/mL) was observed. Similar concentration dependences were obtained using MADLS. It can be seen that at a lower protein concentration of 0.01 mg/mL, the aggregate size significantly exceeded the cases when there were high protein concentrations at the same amounts of NPs (Figure 6d concentrations points of TSC-IONPs 5 × 10^11^ and 10^12^ mL^−1^). This is because, at high protein concentrations, the nanoparticles are covered with a protein corona and are less likely to aggregate with other NPs. At low protein concentrations, several nanoparticles can interact with one protein molecule at once, which increases the likelihood of aggregation.

The appearance of aggregates is also accompanied by changes in the ζ-potential and pH of the solution (Table 1). These changes are expected. The addition of positively charged proteins (4.1 ± 1.3 mV for 0.01 mg/mL, 21.1 ± 2.7 mV for 0.4 mg/mL, and 26.8 ± 4.9 mV for 5 mg/mL HEWL) to negatively charged nanoparticles (−32.1 ± 2.3 mV for TCS-IONPs 10^13^ mL^−1^) led to the formation of aggregates with a charge close to zero for high protein concentrations (2.6 ± 0.2 mV for 0.4 mg/mL and 10.1 ± 0.5 mV for 5 mg/mL HEWL).

The observed protein aggregation upon the addition of NPs does not lead to protein denaturation. This conclusion can be drawn from the data on the activity of HEWL and also confirms the fact that we can release single molecules with the help of shaking. The activity of HEWL after the addition of aggregates to *Micrococcus* did not change even after a daily content in a solution with TSC-IONPs (see Appendix A). A denatured protein cannot quickly regain its shape. Previously, chemically (GdnHCL 6 M and DTT 50 mM) denatured HEWL, after removing the denaturing conditions, was restored within tens of minutes [50]. Similar effects were observed for other biomolecules [66]. Additionally, this conclusion is confirmed by experiments with a vortex. During the vibration processing of the solution, the appearance of a peak of single HEWL molecules from the solution with aggregates was observed (Figure 7). Without a significant change in the native structure, the coagulation effect with NPs is used to precipitate various proteins. For example, iron oxide nanoparticles coated with sodium citrate were used as a coagulating agent for protein precipitation [6].

It is difficult to determine the toxicity of IONPs and their mechanisms. The Fenton reaction may be the main cause of cytotoxicity for uncovered IONPs for eukaryotic [67] and prokaryotic cells [68]. For large iron nanoparticles >40 nm in size, which may already be ferromagnetic, the danger to the body can be associated with the formation of aggregates of nanoparticles because of their magnetic moment. Another possible mechanism is rather strong magnetic fields up to 100 mT near the NP surface [52]. Such fields can significantly change the rate of some magnetically dependent chemical reactions [69], which can also increase cytotoxicity.

The last possible mechanism is aggregation with proteins. Such a mechanism is taking place most likely for small <40 nm coated NPs, whose molecules can bind to proteins. In 10 nm TSC-IONPs at a concentration of 10^13^, the calculated concentration of Fe was approximately ~0.1 mM. However such concentrations of IONPs in other studies led to a drop in cell viability by only 10–20% for IONPs both coated with TSC [39] and uncoated [70]. This may be due to differences in the level of cytotoxicity of particles of different sizes. It is known that smaller NPs give a more pronounced cytotoxic effect. In a study with 7.3 nm, 15.1 nm, and 30.0 nm NPs at the same concentrations, the smallest particles produced more ROS in mitochondria and eventually damaged them [70]. However, this effect is probably simply related to an increase in the surface area of iron oxide. On the other hand, the small particle size increases the adhesive properties of NPs [71] and can lead to the formation of large aggregates with proteins. Based on the results of this study with both proteins and cells, it can be the main cause of cytotoxicity for small ~3 nm TSC-IONPs.

In this work, we showed similar concentration dependences of the obtained TSC-IONPs for cytotoxicity and protein aggregation. Of course, our conclusions about the causes and mechanisms of NP cellular toxicity are incomplete and require further research.

## 5. Conclusions

In this paper, iron oxide nanoparticles coated with trisodium citrate were obtained. Nanoparticles self-assembling stable clusters were ~10 and 50–80 nm in size, consisting of NPs 3 nm in size. It has been shown that the addition of HEWL in a wide range of concentrations to clusters consisting of nanoparticles leads to the decomposition of clusters into individual nanoparticles. At the same time, individual nanoparticles can form aggregates in HEWL molecules not according to the “crown” type, but according to the “bun with raisin” type. It is shown that individual protein molecules can be isolated from the formed aggregates under mechanical action.

After the destruction, the nanoparticles participate in the formation of large aggregates whose sizes depend on the concentration of the protein and nanoparticles, and the aggregates can reach the size of a micron. Such aggregation was observed by several methods: MADLS, optical absorption, fluorescence, TEM, and optical microscopy.

It is important to note that the concentrations of NPs at which the protein aggregation was also toxic to cells. There was a sharp decrease in the survival of mouse fibroblasts (Fe concentration ~75–100 μM), while the ratio of apoptotic to all dead cells increased. In addition, at low concentrations of NPs, an increase in cell size was observed.

## Figures and Tables

**Figure 1 nanomaterials-12-03960-f001:**
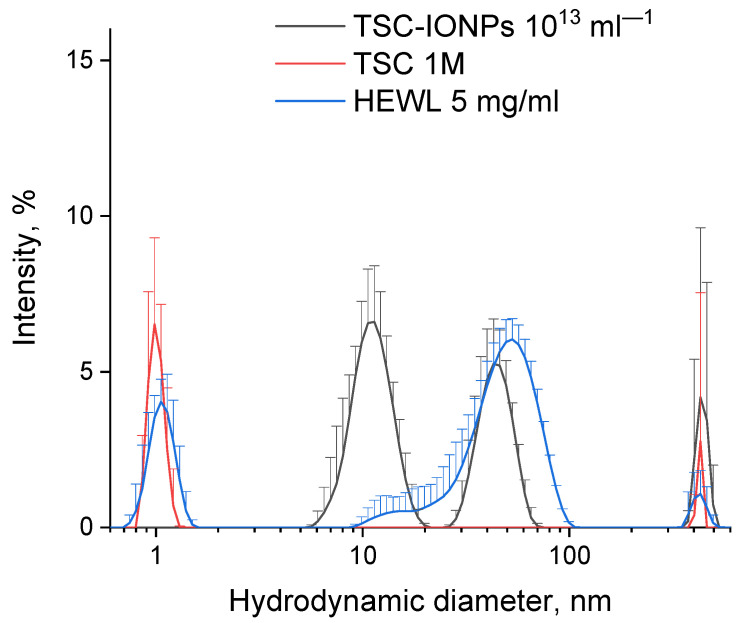
Distribution (weighted by intensity) of hydrodynamic diameters of molecules, clusters, and aggregates for TSC-IONPs 10^13^ mL^−1^, TSC 1 M, and HEWL (5 mg/mL) solution without NPs and TSC. Polydispersity index for TSC-IONPs 0.34 ± 0.15 (mean ± SD), for TSC 0.68 ± 0.16, and for HEWL 1.03 ± 0.17.

**Figure 2 nanomaterials-12-03960-f002:**
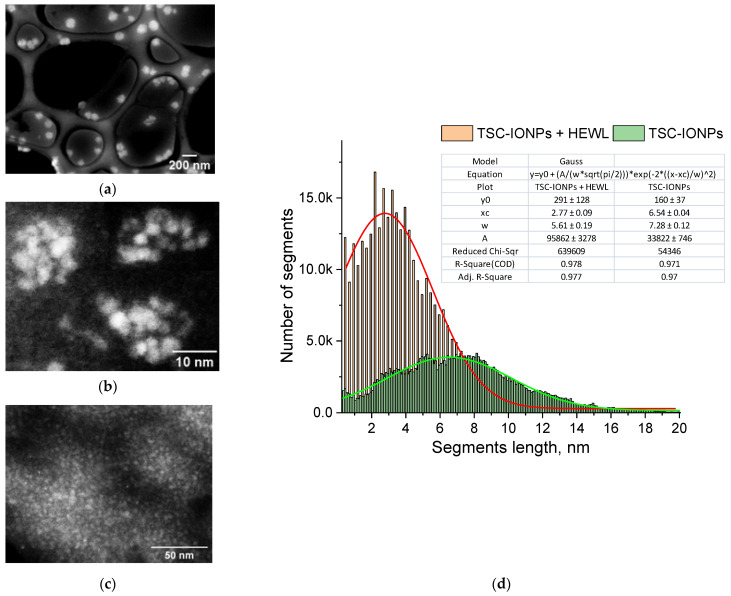
TEM-images: (**a**) large clusters of TSC-IONP nanoparticles, (**b**) small clusters of TSC-IONPs, (**c**) TSC-IONPs (10^13^ mL^−1^) with HEWL (5 mg/mL), (**d**) the distribution of the lengths of the segments connecting the boundary pixels of the NPs in the TEM images (**b**,**c**). The lines show the Gaussian approximation of the distributions of segments for NPs in the clusters (green line, (**b**)) and NPs in the aggregates with HEWL (red line, (**c**)).

**Figure 3 nanomaterials-12-03960-f003:**
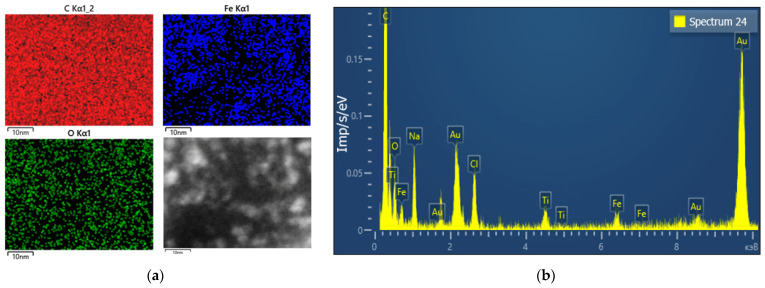
(**a**) EDS analysis of distribution of carbon (left top), oxygen (left bottom), iron (right top) in the sample (right bottom) TSC-IONPs (10^13^ mL^−1^) with HEWL (5 mg/mL); (**b**) energy dispersive spectroscopy results for TSC-IONPs with HEWL.

**Figure 4 nanomaterials-12-03960-f004:**
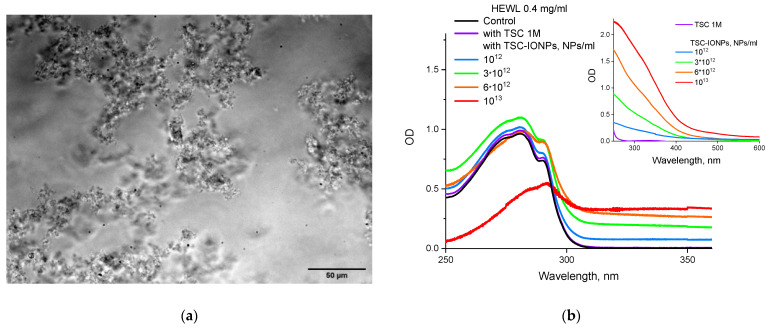
(**a**) Photo of aggregates of HEWL (5 mg/mL) and TSC-IONPs (10^13^ mL^−1^) made with a microscope. (**b**) Absorption of HEWL solution (0.4 mg/mL) with the addition of various concentrations of TSC-IONPs (up to 10^13^ mL^−1^—red line) and TSC 1 M (purple line). Measurements were made relative to water for protein, or relative to the respective concentrations of TSC-IONPs and TSC. The top right inset shows absorption measurements of TSC-IONPs and TSC 1 M solutions concerning water.

**Figure 6 nanomaterials-12-03960-f006:**
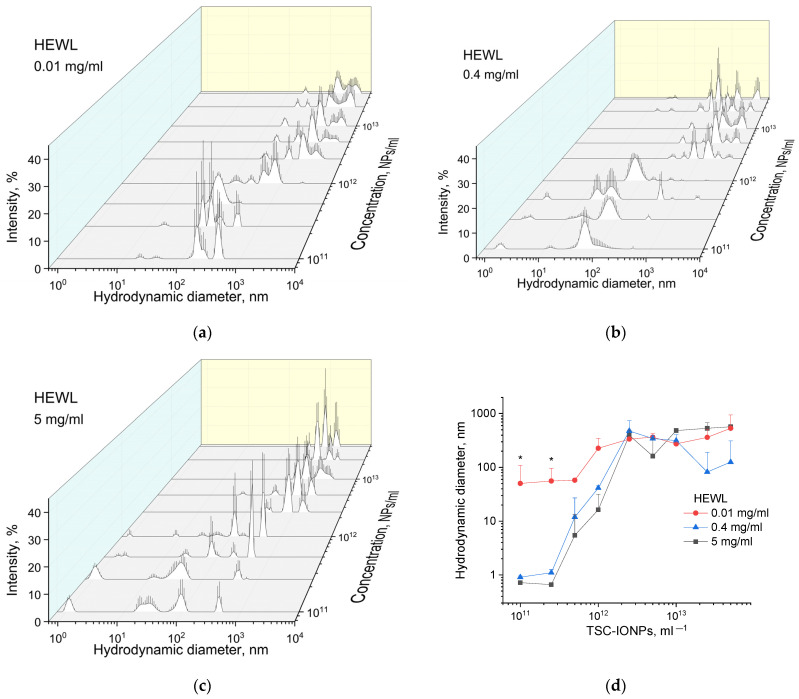
Distribution (weighted by intensity) of hydrodynamic diameters of molecules and aggregates for different concentrations of NPs (from 10^11^ to 5 × 10^13^ mL^−1^) and different concentrations of HEWL: 0.01 (**a**), 0.4 (**b**), 5 (**c**) mg/mL. (**d**) Dependence of the average size of hydrodynamic diameter on the concentration of NPs (* *p* < 0.05). Polydispersity index for all measurements in the Appendix A. Results are presented as mean and SD for five independent experiments.

**Figure 7 nanomaterials-12-03960-f007:**
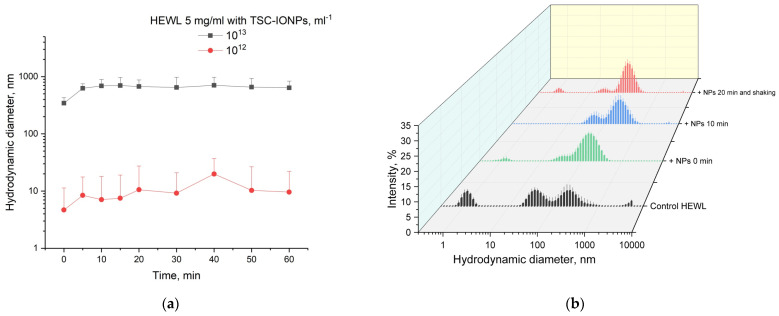
(**a**) Time dependence of the average size of hydrodynamic diameter of HEWL (5 mg/mL) for TSC-IONPs (10^12^ and 10^13^ mL^−1^). Results are presented as mean and SD for three independent experiments. (**b**) DLS intensity data for solution HEWL (10 mg/mL) with TSC-IONPs (10^12^ mL^−1^): HEWL control, immediately after the addition of NPs, after 10 min, after 20 min; in addition, the sample was shaken for 10 s on a Biosan V1-plus vortex. Mean and SD were calculated from five measurements.

**Figure 8 nanomaterials-12-03960-f008:**
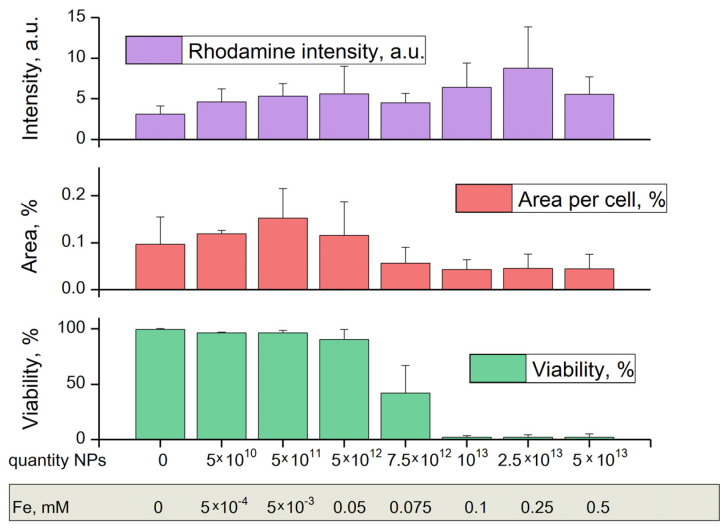
Rhodamine intensity in a fibroblast cell, the area occupied by a fibroblast cell relative to the entire frame area, and fibroblast cell viability after 24 h of exposure to TSC-IONPs of various concentrations. Results are presented as mean and SD for three independent experiments.

**Figure 9 nanomaterials-12-03960-f009:**
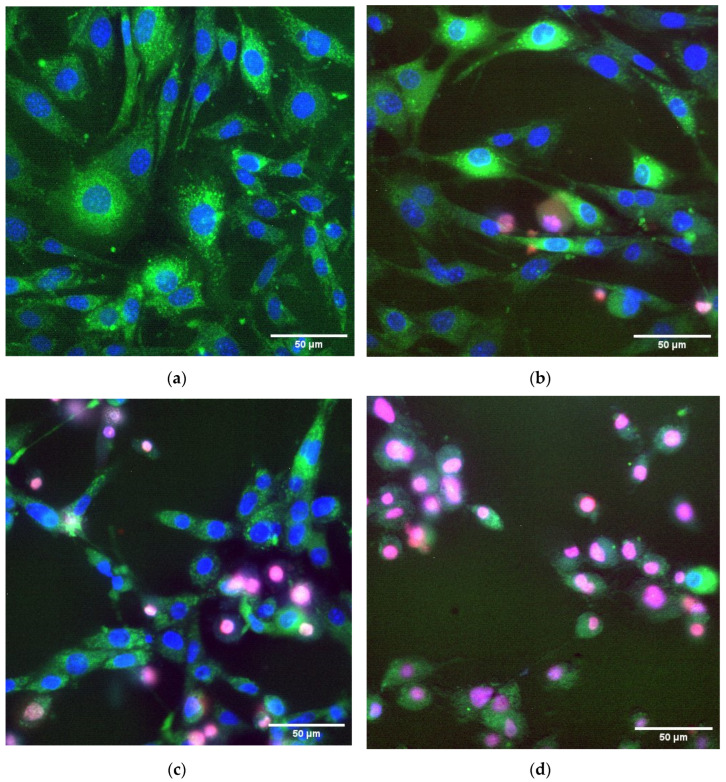
Representative images showing fibroblast cells at various concentrations of addition of iron oxide nanoparticles 24 h after exposure: (**a**) control cells, (**b**) cells with TSC-IONPs 5 × 10^12^ mL^−1^, (**c**) cells with TSC-IONPs 7.5 × 10^12^ mL^−1^, (**d**) cells with TSC-IONPs 10^13^ mL^−1^. Staining with three types of fluorescent dyes: Hoechst 33342 (blue, stains the live cell nucleus), cell nuclei PI (red, stains the nucleus of dead cells), rhodamine-123 (green, mitochondrial membrane potential indicator).

**Figure 10 nanomaterials-12-03960-f010:**
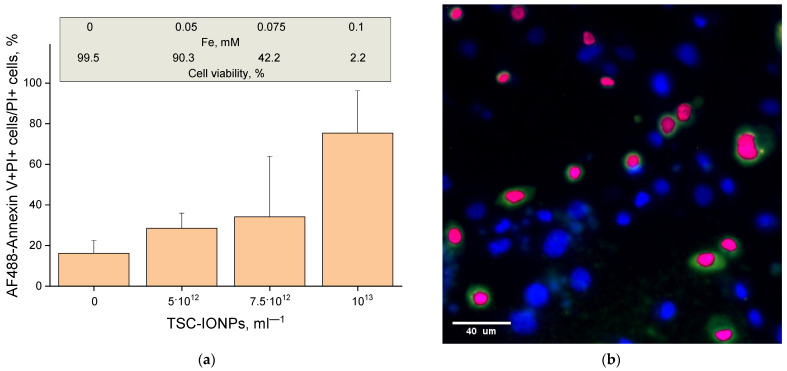
The ratio of apoptotic cells (colored green with AF488-Annexin V+PI+) to the total number of dead cells (colored purple PI+) (**a**), a sample photo of the concentration of TSC-IONPs 7.5 × 10^12^ mL^−1^ (**b**).

**Figure 11 nanomaterials-12-03960-f011:**
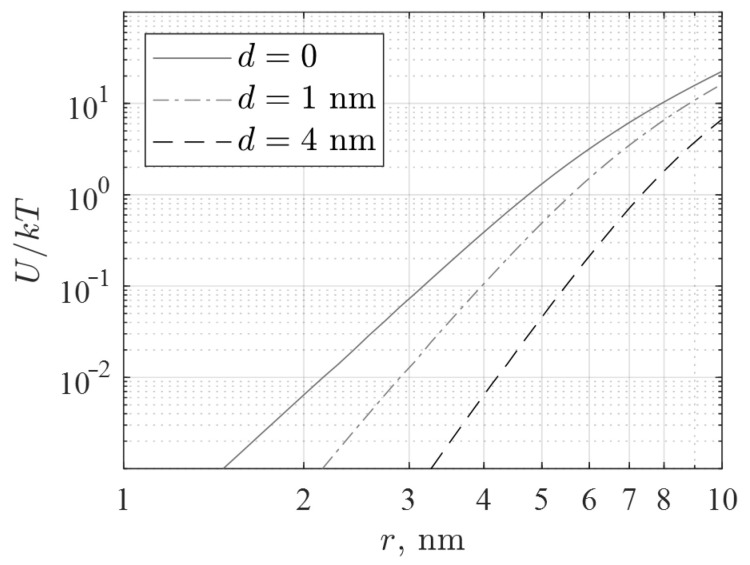
Dependence of the relative value of the magnetic interaction U/kT on the radius r of two contacting IONPs without shells and at different shell thicknesses d: 1 nm for the citrate shell and 4 nm for the protein shell.

**Table 2 nanomaterials-12-03960-t002:** ζ-potential (mV), hydrodynamic diameters of the first peaks (MADLS data), and pH (electrode InLab Expert Pro-ISM measured in SevenExcellence pH/Cond meter S470) for HEWL and TSC-IONPs (10^13^ NPs/mL) and the association. Results are presented as mean and SD for three independent experiments.

Solution	ζ-Potential, mV	Hydrodynamic Diameters of the First Peaks, nm	pH
TSC-IONPs 10^13^ NPs/mL	−32.1 ± 2.3	10.9 ± 0.6	8.6 ± 0.1
HEWL 0.01 mg/mL	4.1 ± 1.3	–	5.4 ± 0.2
HEWL 0.4 mg/mL	21.1 ± 2.7	1.2 ± 0.2	4.1 ± 0.1
HEWL 5 mg/mL	26.8 ± 4.9	1.1 ± 0.1	3.3 ± 0.1
HEWL 0.01 mg/mL + NPs	−25.2 ± 0.8	246 ± 12.3	8.3 ± 0.2
HEWL 0.4 mg/mL + NPs	2.6 ± 0.2	264 ± 13.2	8.1 ± 0.2
HEWL 5 mg/mL + NPs	10.1 ± 0.5	462 ± 11.2	4.8 ± 0.1

## Data Availability

Data are available from the authors upon request.

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
