# Peer review of "Investigation of Aggregation and Disaggregation of Self-Assembling Nano-Sized Clusters Consisting of Individual Iron Oxide Nanoparticles upon Interaction with HEWL Protein Molecules"

_nanomaterials, 2022, doi:10.3390/nano12223960_

Round 1

Reviewer 1 Report

The authors prepared TSC-IONPs with small size. Then the interaction between NPs with HEWL, and the toxicity of NPs were investigated. I think it is unacceptable since data presented are preliminary.

1. What’s the scientific problem do the authors aim to solve?

2, Since the isoelectric points (pI) of HEWL is 10.7, the protein is positive under physiological conditions, which will neutralize the negative charge of TSC-IONPs and lead to the aggregation of NPs. what’s the significance for the authors to choose HEWL to do this work?

3. It is not enough to conclude that the cells undergo apoptosis or necrosis based on the morphology of cells, the style of cell death should be confirmed by further analysis, for example Annexin V/PI staining.

4. Figure 1 is not clear enough. Figure 4b, which sample does the red curve represent?

5. Line 306-320 and Line 330-343 are the same.

Author Response

We are grateful to the reviewer for his work, as well as for valuable comments that will improve our manuscript. The manuscript was proofread by a native speaker. The manuscript contains some of the changes after proofreading. They are labeled yellow.

Comment 1.1

  1. What’s the scientific problem do the authors aim to solve?

Reply 1.1

We have changed the title, abstract, introduction, and conclusion. Part of the data has been moved from Supplementary Materials to the main article. We have more clearly formulated the goals based on the novelty of the relevance of the data we received. 

New title

Investigation of aggregation and disaggregation of self-assembling nano-sized clusters consisting of individual iron oxide nanoparticles upon interaction with HEWL protein molecules.

Added to Abstract

“In this paper, iron oxide nanoparticles coated with trisodium citrate were obtained. Nanoparticles self-assembling stable clusters ~10 and 50–80 nm in size, consisting of NPs 3 nm in size. The stability was controlled by using multi-angle dynamic light scattering and the zeta potential, which was -32±2 mV. Clusters from TSC-IONPs can be destroyed when interacting with a hen egg-white lysozyme. After the destruction of the nanoparticles and proteins are quickly within 5-10 minutes formed aggregates. Their sizes of them depend on the concentration of the lysozyme and nanoparticles and can reach micron sizes. It is shown that individual protein molecules can be isolated from the formed aggregates under shaking. Such aggregation was observed by several methods: multi-angle dynamic light scattering, optical absorption, fluorescence spectroscopy, TEM, and optical microscopy. It is important to note that the concentrations of NPs at which the protein aggregation was also toxic to cells. There was a sharp decrease in the survival of mouse fibroblasts (Fe concentration ~ 75–100 μM), while the ratio of apoptotic to all dead cells increased. Also, at low concentrations of NPs, an increase in cell size was observed.”

Added to introduction

“Another problem is associated with the use of the magnetic properties of NPs. It is known that nanoparticles smaller than 20 nm are superparamagnetic, which is convenient for biomedical applications since they have low residual magnetization without the application of an external magnetic field [17]. However, such NPs are extremely unstable and tend to aggregate[18]. The use of sodium citrate makes it possible to stabilize the aggregation of particles, at least until the formation of long-lived clusters. However, it is not known how stable these clusters will be when interacting with proteins. Especially with positively charged proteins, since sodium citrate has a negative charge.

In this study, a technology was created for obtaining self-assembling clusters, stable in aqueous solutions, consisting of individual iron oxide nanoparticles coated with citrate anions. The paper aimed to study the aggregation and disaggregation of self-assembling nanosized clusters consisting of individual TSC-IONPs upon interaction with HEWL protein molecules.”

Added to conclusion

“In this paper, iron oxide nanoparticles coated with trisodium citrate were obtained. Nanoparticles self-assembling stable clusters ~10 and 50–80 nm in size, consisting of NPs 3 nm in size. It has been shown that the addition of HEWL in a wide range of concentrations to clusters consisting of nanoparticles leads to the decomposition of clusters into individual nanoparticles. At the same time, individual nanoparticles can form aggregates in HEWL molecules not according to the “crown” type, but according to the “bun with raisin” type. It is shown that individual protein molecules can be isolated from the formed aggregates under mechanical action.

After the destruction, the nanoparticles participate in the formation of large aggregates whose sizes depend on the concentration of the protein and nanoparticles and the aggregates can reach the size of a micron. Such aggregation was observed by several methods: MADLS, optical absorption, fluorescence, TEM, and optical microscopy.

It is important to note that the concentrations of NPs at which the protein aggregation was also toxic to cells. There was a sharp decrease in the survival of mouse fibroblasts (Fe concentration ~ 75–100 μM), while the ratio of apoptotic to all dead cells increased. Also, at low concentrations of NPs, an increase in cell size was observed.”

Comment 1.2

Since the isoelectric points (pI) of HEWL is 10.7, the protein is positive under physiological conditions, which will neutralize the negative charge of TSC-IONPs and lead to the aggregation of NPs. what’s the significance for the authors to choose HEWL to do this work?

Reply 1.2

A positively charged protein was chosen to study the stability of nanoparticle clusters. It was interesting to follow whether a protein crown of clusters would form or whether the interaction with the protein would be different. We also experimented with negatively charged proteins, such as BSA. We have not seen micron-sized aggregates.

Comment 1.3

It is not enough to conclude that the cells undergo apoptosis or necrosis based on the morphology of cells, the style of cell death should be confirmed by further analysis, for example Annexin V/PI staining.

Reply 1.3

We did tests with apoptosis and added the results to the text of the article. As a result, almost three-quarters of all died cells at a concentration of NPs of 1013 were apoptotic.

Comment 1.4

Figure 1 is not clear enough. Figure 4b, which sample does the red curve represent?

Reply 1.4

We rewrote the captions for both figures. Red line on the Figure 4b is absorption of HEWL solution (0.4 mg/ml) with the addition 1013 ml-1 TSC-IONPs (up to). Measurements were made relative to the respective concentrations of TSC-IONPs. We have swapped figures 1 and 2 in the text.

Comment 1.5

Line 306-320 and Line 330-343 are the same.

Reply 1.5

Removed duplicate text

Reviewer 2 Report

Manuscript Number: nanomaterials-1954122   

Interaction of Iron Oxide Nanoparticles Coated with Sodium Citrate with Hen Egg-White Lysozyme 

This is a creative article in the means of assays used to study the interaction of HEWL with nanoparticles and their cytotoxicity. The manuscript's English usage has many problems and could use some editing to clear up word usage, punctuation and grammar. Many sentences were unclear. I had to come back after the first read to focus on science. The manuscript may be considered only once they extensively edited English (use shorter sentences and check the grammar thoroughly) and include the FTIR experiment. The experiments appear to be technically sound, but there is a need for some rewriting to clarify the different features of the study.

1.     The TSC-IONPs prepared in this work form two populations: 50 – 100 nm and ~10 nm. How were TEM images of each population prepared? It is not clear if both populations dissociate in the presence of protein or only the smaller clusters of 10 nm size. Also, I suggest deleting the approximate sizes of NPs from the second sentence (line 229), as the sizes obtained from the TEM image analysis are mentioned later in the text.

2.     Using the term "aggregates" for both nanoparticles and samples of nanoparticles with protein is confusing for readers. I suggest that the authors should use clusters for nanoparticles only and aggregates for nanoparticles with protein.

3.     In the Results section (lines 227 – 242), clarify which part of Figure 1 you are referring to as Fig. 1(c) and (b) do not contain a green or red line, but Fig. 1 (d) does. This figure is not mentioned at all within the text.

4.     Line 244: one comma is missing.

5.     Specify in Fig. 2 caption or description within the manuscript whether the blue line represents only protein or is a mixture of protein and nanoparticles. If it is only a HEWL solution, what is your explanation for the presence of large aggregates between 10 and 100 nm in size? Is HEWL aggregating in the solution? Moreover, these results indicate that monomeric HEWL is only ~ 1 nm in size, but several other papers showed that the hydrodynamic diameter of ~14 kDa HEWL is 2 - 3 nm in neutral conditions.

6.     The statement "The formation of large micron protein aggregates upon the addition of sufficiently high concentrations of TSC-IONPs is confirmed both with a fluorescent microscope (Fig. 4(a)) and with absorption on a spectrophotometer (Fig. 4(b))." is not correct. The size of aggregates cannot be determined using UV-VIS spectroscopy but by MADLS (Fig. 6). The absorption and fluorescence spectra (Fig. 4b and 5) indicate the binding of protein molecules to the nanoparticle surface.

7.     It is rather strange to observe almost the same concentration dependencies for the increasing concentration of HEWL (Fig. 5 (b)). Add an explanation of the observed behavior for HEWL concentrations of 0.4 mg/ml and higher. Can authors describe in greater detail how the curves were normalized?

8.     Line 262: it is exited, not excitated.

9.     Based on the protein activity assay, the authors state that HEWL "does not significantly change its native conformation upon formation of aggregates with NPs or quickly returns to it…". It might be accurate, but HEWL can maintain its biological activity even in non-native conformation. The authors should support this statement with a more direct method for structural protein characterization (FTIR).

10.  Sections 306 – 319 and 330 – 343 are identical.

11.  Line 387: "It may seem that the formation of aggregates is due to a change in pH from 387 4 to 8." It is not clear what kind of aggregates do authors describing. In the case of HEWL, the pH changes from 4 to 8 only for HEWL concentration of 0.4 mg/ml. In the case of 5 mg/ml HEWL concentration, the pH changes from 3.3 to 4.8.

12.  I would suggest including FTIR spectroscopy to confirm the presence of a TSC shell on the IONPs surface; as you mention, "...using EDS-analysis it is very difficult 467 to show that an element is on the shell, and not in the solution. "(lines 465 – 473).

13.  The identical curves of fluorescence quenching in the presence of different HEWL concentrations need more explanation (Fig. 5 (b)) in the discussion part. The aggregation of protein can be plausible, but this hypothesis needs to be supported by another method, i.e. turbidity. How does the turbidity of HEWL change with increasing concentration of NPs?

14.  I would move the part of HEWL with NPs aggregates cytotoxicity from the discussion part to the results. 

Author Response

We are grateful to the reviewer for his work, as well as for valuable comments that will improve our manuscript. We have changed the title, abstract, introduction, and conclusion. Part of the data has been moved from Supplementary Materials to the main article. We have more clearly formulated the goals based on the novelty of the relevance of the data we received.  The manuscript was proofread by a native speaker. The manuscript contains some of the changes after proofreading (labeled yellow).

Comment 2.1

a)The TSC-IONPs prepared in this work form two populations: 50 – 100 nm and ~10 nm. How were TEM images of each population prepared? b)It is not clear if both populations dissociate in the presence of protein or only the smaller clusters of 10 nm size. c) Also, I suggest deleting the approximate sizes of NPs from the second sentence (line 229), as the sizes obtained from the TEM image analysis are mentioned later in the text.

Reply 2.1

  1. a) We are grateful to the reviewer for his work, as well as for valuable comments that will improve our manuscript. All samples for NPs and NPs with protein solution were prepared for TEM as follows. Approximately 0.25 μl drop of solution was applied to a round gold mesh ~4 mm in diameter. The samples were dried at room temperature for 10 min and evacuated. Sample preparation clarifications added to Materials and Methods. The sample on which photographs were taken with clusters of 50 - 100 nm and ~10 nm was the same (in Supplementary materials, see Figure Fig. S1a, a large area of ​​​​the sample is shown). Different sizes of clusters were photographed at different magnifications and were typical for the entire surface.

b)After the addition of lysozyme, all clusters, both small and large, disintegrated into single nanoparticles that aggregated with the protein.

  1. c) Done

We changed the text in the results This figure shows the distribution of segments between all points of the boundaries of the NPs.to another text “After the addition of lysozyme, no individual NP clusters were seen in the samples (see figures Fig. S1b, Fig. S1c, and Fig. S1d in Supplementary materials). Figure 1(c) shows the distribution of segments between all points of the boundaries of the NPs.”

We changed the text in the results “Images obtained with a transmission microscope suggest that synthesized TSC-IONPs have a spherical shape with ~3 nm size and they form big (~50-100 nm) and small (~10 nm) aggregates” to another text “Images obtained with a transmission microscope suggest that synthesized TSC-IONPs have a spherical shape with ~3 nm size and they form big and small clusters”

Comment 2.2

Using the term "aggregates" for both nanoparticles and samples of nanoparticles with protein is confusing for readers. I suggest that the authors should use clusters for nanoparticles only and aggregates for nanoparticles with protein.

Reply 2.2

This is a very valuable note. We have made changes to the text. Now everything related to the aggregation of the nanoparticles themselves is called "clusters"

Comment 2.3

In the Results section (lines 227 – 242), clarify which part of Figure 1 you are referring to as Fig. 1(c) and (b) do not contain a green or red line, but Fig. 1 (d) does. This figure is not mentioned at all within the text.

Reply 2.3

We have swapped figures 1 and 2 in the text.

We changed the text “It can be seen from the distribution that the most probable size of the NPs before interaction with the protein (Fig. 1(b), green line) lies in the region of 6.5-8 nm. When interacting with lysozyme (Fig. 1(c), red line), the most probable size of the NPs becomes 2.8 nm.”  to another text “It can be seen from the distribution that the most probable size of the NPs before interaction with the protein (Fig. 2(b) and respectively green line in Fig. 2(d)) lies in the region of 6.5-8 nm. When interacting with lysozyme (Fig. 2(c) and respectively red line in Fig. 2(d)), the most probable size of the NPs becomes 2.8 nm.”

Comment 2.4

Line 244: one comma is missing.

Reply 2.4

Done

Comment 2.5

  1. a) Specify in Fig. 2 caption or description within the manuscript whether the blue line represents only protein or is a mixture of protein and nanoparticles.
  2. b) If it is only a HEWL solution, what is your explanation for the presence of large aggregates between 10 and 100 nm in size? Is HEWL aggregating in the solution?
  3. c) Moreover, these results indicate that monomeric HEWL is only ~ 1 nm in size, but several other papers showed that the hydrodynamic diameter of ~14 kDa HEWL is 2 - 3 nm in neutral conditions.

Reply 2.5

  1. a) We changed the text in the caption HEWL (5 mg/ml) to another text “HEWL (5 mg/ml) solution without NPs and TSC”
  2. b) Yes, these may be separate aggregates of lysozyme, but their number is ~108-109 times less than the native protein. In addition, it can be bubbles of dissolved gases that are present in water [Penkov, N.V. Temporal Dynamics of the Scattering Properties of Deionized Water. Physics of Wave Phenomena 2020, 28, 135-139, doi:10.3103/S1541308X20020132].
  3. c) We changed the textIt is known that the hydrodynamic diameter of proteins increases with an increasing salt concentration in the buffer [42]. In our previous work, the size of HEWL in 50mM Tris-HСl was 3.85 nm [43]” to another text “The hydrodynamic diameter of lysozyme measured using DLS under control conditions turned out to be ~1.2 nm (Fig. 1). This is an interesting phenomenon that I think many researchers working with the DLS method have encountered. Measurement DLS the hydrodynamic diameter of macromolecules is strongly dependent on the solvent and this is taking into account corrections for viscosity and refractive index necessary to fine particle sizes from the autocorrelation function. For example, the lysozyme we measured earlier in 50 mM Tris-HCl (pH = 8.0) had a size of 3.8 nm [51]. Or, in this work, the hydrodynamic diameter of lysozyme increased with the addition of TSC and reached 2.5 nm at 1 mM TSC solution (Fig. S5 Supplementary Materials). Other authors have also observed particle sizes in DLS measurements for proteins [50] and even for carboxylated latex beads [44]. Moreover, in the latter case, the differences in particle sizes between different solvents reached one order of magnitude.”

Comment 2.6

 The statement "The formation of large micron protein aggregates upon the addition of sufficiently high concentrations of TSC-IONPs is confirmed both with a fluorescent microscope (Fig. 4(a)) and with absorption on a spectrophotometer (Fig. 4(b))." is not correct. The size of aggregates cannot be determined using UV-VIS spectroscopy but by MADLS (Fig. 6). The absorption and fluorescence spectra (Fig. 4b and 5) indicate the binding of protein molecules to the nanoparticle surface.

Reply 2.6

We do not quite agree with the reviewer's statement. Of course, the results of fluorescence are not as unambiguous as DLS, however, shifts in the maxima or the appearance of new peaks may indicate a change in the native conformation of the protein. With the formation of aggregates, the absorption increases both in the visible and in the UV region, and as a result, the decreases reduce peaks of the maxima in fluorescence spectra. For example In our previous work [Sarimov, R.M.; Binhi, V.N.; Matveeva, T.A.; Penkov, N.V.; Gudkov, S.V. Unfolding and aggregation of lysozyme under the combined action of dithiothreitol and guanidine hydrochloride: Optical studies. International Journal of Molecular Sciences 2021, 22, 1-16, doi:10.3390/ijms22052710], we added DTT to lysozyme (S–S bonds were broken) and saw both an increase in aggregation of lysozyme in DLS and an increase in absorption in the UV region. In the case of denaturation of lysozyme with the addition of GdnCl, a slight increase in the hydrodynamic diameter (from 3.8 nm to 5.7 nm) was accompanied by a slight increase in absorption.

Comment 2.7

It is rather strange to observe almost the same concentration dependencies for the increasing concentration of HEWL (Fig. 5 (b)). Add an explanation of the observed behavior for HEWL concentrations of 0.4 mg/ml and higher. Can authors describe in greater detail how the curves were normalized?

Reply 2.7

We have added fluorescence spectra to the manuscript (Fig. 5(c)-5(f)), which normalized the data, Table 1 with the location of the emission maxima for these spectra, as well as Table S3 with settings in the supplementary materials.

Added texts to Materials and Methods: “The normalization concentration dependence (Figure 5(b)) according to fluorescent data (Figures 5(c)-5(f)) was constructed as follows. All samples, both control and with nanoparticles, were excited at a wavelength of 300 nm and the emission spectra were recorded. The maxima for controls of different concentrations were approximately at the emission wavelength of 337 nm (Table 1). The maxima for a solution of proteins with NPs were in the region of 336–337, and at high concentrations of NPs, they shifted to the region above 340 nm (Table 1) in the emission spectrum. Next, we found the maximum intensity for each NP concentration and normalized it to the intensity of the control maximum.”

Added texts about original fluorescence spectra to Results: “The emission of HEWL samples of various concentrations upon the addition of NPs in various concentrations is shown in figures 5(b)-5(f). The samples were excited at 300 nm, the emission peak for different concentrations of lysozyme in the control was ~337 nm, and when TSC-IONPs were added, it shifted to the long wavelength region (Fig. 5(c)-5(f), Tab. 1). For comparison, Figure 5(b) shows normalized data, where the intensities of fluorescence maxima for each concentration of NPs and HEWL were normalized to the intensities of the corresponding control. The same fluorescent concentration curves for proteins 0.4, 5, and 100 mg/ml indicate that the decrease in fluorescence is primarily associated with the addition of NPs, which absorb well in this range (see the top inset in Figure 4(c)). But the change in the course of the curve at a protein concentration of 0.01 mg/mL indicates the interaction of protein NPs.”

and

“In addition, we observed a shift in the fluorescence maximum upon the addition of NPs (Fig. 5(c)-5(f), Tab. 1). Characteristically, this shift depends on the protein concentration (Tab. 1). This indicates that the protein interacts with nanoparticles to form aggregates.”

and Discussion: ” Particularly interesting data were obtained when measuring fluorescence. When TSC-IONPs are added to the protein solution, the fluorescence decreases (Fig. 5). The decrease in fluorescence, in our opinion, occurs for two reasons. First, the NPs themselves absorb well in the range of 260–300 nm (see the inset at the top right of Fig. 4(b)). However, this absorption is not enough to explain the dependence of the fluorescence decay on the protein concentration (Fig. 5(b)). For example, for a concentration of NPs of 1012 ml-1, the decrease in fluorescence for a HEWL concentration of 0.4 mg/ml and above is about 20%, while for a concentration of 0.01 mg/ml it is already 50%. In addition, this does not explain the shift in the fluorescence maximum that is observed when NPs are added (Fig. 5, Table 1). Moreover, this shift is the greater, the greater the concentration of lysozyme. Such shifts in fluorescence were previously recorded during denaturation, both thermally [63] and with guanidine hydrochloride [64]. Shifts in fluorescence and the appearance of new peaks in the long-wavelength region were also observed during the formation of amyloid fibrils [65]. The second and main reason, in our opinion, is protein aggregation, as a result of which less excitatory light reaches fluorescent amino acids (Trp, Tyr, Phe) and absorption of emission light (330–360 nm) by the protein aggregates also occurs (Fig. 5(b)- 5(f)).

We repeated the experiment with the same fluorimeter settings for all samples, made sure that the results repeated the original data (Figure 5(b)) and don't depend on the settings, and added the results to Supplementary Materials (Figures S12, S13, S14, and Table S4).

Comment 2.8

Line 262: it is exited, not excitated.

Reply 2.8

Done

Comment 2.9

Based on the protein activity assay, the authors state that HEWL "does not significantly change its native conformation upon formation of aggregates with NPs or quickly returns to it…". It might be accurate, but HEWL can maintain its biological activity even in non-native conformation. The authors should support this statement with a more direct method for structural protein characterization (FTIR).

Reply 2.9

We have taken FTIR measurements and added the description and results to the Supplementary Materials (Fig. S16). Unfortunately, we did not obtain significant differences between the peaks of the control protein and the NP protein. Perhaps this is due to the insufficient sensitivity of our device (we had a single-beam spectrometer). In addition, for FTIR, as well as for TEM, there are significant limitations in the study of the native state of biomolecules in solutions. Since the samples are dried before measurement, the state of the molecules in the dried sample will strongly depend on the sample preparation process and may be very far from the actual native conformation in the solution.

Comment 2.10

Sections 306 – 319 and 330 – 343 are identical.

Reply 2.10

Removed duplicate text

Comment 2.11

Line 387: "It may seem that the formation of aggregates is due to a change in pH from 387 4 to 8." It is not clear what kind of aggregates do authors describing. In the case of HEWL, the pH changes from 4 to 8 only for HEWL concentration of 0.4 mg/ml. In the case of 5 mg/ml HEWL concentration, the pH changes from 3.3 to 4.8

Reply 2.11

We have carried out additional experiments with the addition of NaOH to HEWL (5 mg/ml). In this case, the pH was brought up to almost 12. We didn’t observe the formation of large aggregates up to pH=11.8 (see Fig. S17 in Supplementary Materials). The only noticeable change is the shift of the first peak. We discuss in the text that such a shift is possible and has been observed previously with the addition of salts.

Comment 2.12

 I would suggest including FTIR spectroscopy to confirm the presence of a TSC shell on the IONPs surface; as you mention, "...using EDS-analysis it is very difficult 467 to show that an element is on the shell, and not in the solution. "(lines 465 – 473).

Reply 2.12

See Reply 2.9

Comment 2.13

 The identical curves of fluorescence quenching in the presence of different HEWL concentrations need more explanation (Fig. 5 (b)) in the discussion part. The aggregation of protein can be plausible, but this hypothesis needs to be supported by another method, i.e. turbidity. How does the turbidity of HEWL change with increasing concentration of NPs?

Reply 2.13

We have added protein with NPs absorption data near UV and visible regions in Supplementary materials (Fig. S15). When NPs are added, the OD increases, which indicates both an increase in absorption and scattering. The strange behavior of the curve at a concentration of 1013 ml-1 probably indicates a redistribution of large aggregates in the volume of the cuvette.

Comment 2.14

I would move the part of HEWL with NPs aggregates cytotoxicity from the discussion part to the results. 

Reply 2.14

We replaced to the results  “As mentioned above, an increase in the fluorescence intensity of Rhodamine 123 is associated with a decrease in the mitochondrial membrane potential [58]. It is believed that a decrease in the membrane potential is associated with disruption of the functioning of mitochondria and the onset of processes from cell destruction, and mitophagy [59]. In addition, a decrease in the mitochondrial membrane potential, including the one determined by the fluorescence of Rhodamine 123, can be a marker of the early stages of apoptosis [60], which is consistent with data on cell death and the decrease in the area at high concentrations of NPs (Fig. 7, 8). Mean fluorescence intensity can depend on the area of a cell. But in our case differences in Rhodamin-123 fluorescence intensities value between control and IONPs-treated cells cannot be explained only by a change of cell area. Cell area variation is less than fluorescence intensities value variation, therefore there are additional reasons other than the change of cell area. Such a sharp sigmoid curve for cell viability is somewhat unexpected. Previously observed toxic effects by other authors showed a tenfold decrease in survival (from ~90% to ~10%) with a change in the concentration of iron (for IONPs ~ 40 nm) in the solution by two orders of magnitude from 0.1 mM to 10 mM [61]. However it should not be excluded that such a sharp sigmoidal curve is a feature of the cell line.” and “In separate survival experiments, HEWL (0.4 mg/ml, final concentration in cell solution) or TSC (50 mM, final concentration in cell solution) was also added to the TSC-IONPs (1013 ml-1) solution before adding NPs to the cell culture (see in the Supplementary materials). The aim was to check how much the toxicity of the particles would decrease if they were bound to a protein or additional citrate. In other studies, with additional protein coating of NPs, the toxicity of NPs is usually greatly reduced [40]. However, no effect on cell viability was observed. Cell viability was 1.1±0.8% with the addition of NPs and HEWL (0.4 mg/ml) or 3.4±3.1% with the addition of NPs and TSC (50 mM). In addition only TSC-IONPs (1013 ml-1) cell viability was 2.0±0.6%.”

Reviewer 3 Report

In this paper, iron oxide nanoparticles coated with trisodium citrate were obtained and can be destroyed upon interaction with a protein.  The iron nanoparticles aggregation can become toxic for cells. The discussion is clear in this paper and some arears that need improvement.

1. TEM is not clear enough. Do you have clearer pictures?

2. Some graphs can be merged, such as Fig7-9.

3. The language needs further examination.

Author Response

We are grateful to the reviewer for his work, as well as for valuable comments that will improve our manuscript. We have changed the title, abstract, introduction, and conclusion. Part of the data has been moved from Supplementary Materials to the main article. We have more clearly formulated the goals based on the novelty of the relevance of the data we received.  The manuscript was proofread by a native speaker. The manuscript contains some of the changes after proofreading (labeled yellow).

Comment 3.1

TEM is not clear enough. Do you have clearer pictures?

Reply 3.1

Unfortunately, all available pictures are either in the article or in Supplementary Materials.

Comment 3.2

Some graphs can be merged, such as Fig7-9.

Reply 3.2

Done

Comment 3.3

 The language needs further examination.

Reply 3.3

The manuscript was proofread by a native speaker. The manuscript contains some of the changes after proofreading. They are labeled yellow.